# Inactivation of TGFβ receptors in stem cells drives cutaneous squamous cell carcinoma

Patrizia Cammareri[1,*], Aidan M. Rose[2,*], David F. Vincent[1,*], Jun Wang[3], Ai Nagano[3], Silvana Libertini[1], Rachel A. Ridgway[1], Dimitris Athineos[1], Philip J. Coates[4], Angela McHugh[2], Celine Pourreyron[2], Jasbani H.S. Dayal[2], Jonas Larsson[5], Simone Weidlich[6], Lindsay C. Spender[2], Gopal P. Sapkota[6], Karin J. Purdie[7], Charlotte M. Proby[2], Catherine A. Harwood[7], Irene M. Leigh[2,7], Hans Clevers[8], Nick Barker[9], Stefan Karlsson[5], Catrin Pritchard[10], Richard Marais[11], Claude Chelala[3], Andrew P. South[2,12], Owen J. Sansom[1] & Gareth J. Inman[2]

Melanoma patients treated with oncogenic BRAF inhibitors can develop cutaneous squamous cell carcinoma (cSCC) within weeks of treatment, driven by paradoxical RAS/RAF/MAPK pathway activation. Here we identify frequent *TGFBR1* and *TGFBR2* mutations in human vemurafenib-induced skin lesions and in sporadic cSCC. Functional analysis reveals these mutations ablate canonical TGFβ Smad signalling, which is localized to bulge stem cells in both normal human and murine skin. MAPK pathway hyperactivation (through *Braf^{V600E}* or *Kras^{G12D}* knockin) and TGFβ signalling ablation (through *Tgfbr1* deletion) in LGR5[+ve] stem cells enables rapid cSCC development in the mouse. Mutation of *Tp53* (which is commonly mutated in sporadic cSCC) coupled with *Tgfbr1* deletion in LGR5[+ve] cells also results in cSCC development. These findings indicate that LGR5[+ve] stem cells may act as cells of origin for cSCC, and that RAS/RAF/MAPK pathway hyperactivation or *Tp53* mutation, coupled with loss of TGFβ signalling, are driving events of skin tumorigenesis.

[1] Wnt Signaling and Colorectal Cancer Group, Cancer Research UK Beatson Institute, Institute of Cancer Sciences, Glasgow University, Garscube Estate, Switchback Road, Glasgow G61 1BD, UK. [2] Division of Cancer Research, School of Medicine, University of Dundee, Dundee DD1 9SY, UK. [3] Bioinformatics Unit, Barts Cancer Institute, Queen Mary University of London, Charterhouse Square, London EC1M 6BQ, UK. [4] Tayside Tissue Bank, School of Medicine, University of Dundee, Dundee DD1 9SY, UK. [5] Molecular Medicine and Gene Therapy, Lund Strategic Center for Stem Cell Biology, Lund University, Lund 221 00, Sweden. [6] MRC Protein Phosphorylation Unit, School of Life Sciences, University of Dundee, Dundee DD1 5EH, UK. [7] Centre for Cutaneous Research, Barts and the London School of Medicine and Dentistry, Queen Mary University of London, London E1 2AT, UK. [8] Hubrecht Institute, Utrecht 3584 CT, The Netherlands. [9] Institute of Medical Biology, Immunos 138648, Singapore. [10] Department of Biochemistry, University of Leicester, Leicester LE1 9HN, UK. [11] The Paterson Institute for Cancer Research, Manchester M20 4BX, UK. [12] Department of Dermatology and Cutaneous Biology, Thomas Jefferson University, Philadelphia, Pennsylvania 19107, USA. * These authors contributed equally to this work. Correspondence and requests for materials should be addressed to O.J.S. (email: o.sansom@beatson.gla.ac.uk) or to G.J.I. (email: g.j.inman@dundee.ac.uk).

The development of epithelial tumours is generally accepted to take place over several years, involving the accumulation of mutations that drive tumour progression[1]. However, some tumours contain a relatively low mutation burden[2] and develop rapidly, without progression from benign intermediary stages, suggesting a potential stem cell origin[3]. Data from murine model systems illustrate a tumour's ability to form from both stem and differentiated cells. Within intestinal epithelium, loss of *Apc* in the LGR5$^{+ve}$ stem cell compartment leads to adenoma, whereas tumours rarely form from differentiated cells[4]. Conversely, we have shown that targeting *Kras*, in addition to *Apc*, can de-differentiate intestinal villi and permit tumour formation[5]. Thus, the tumour cell of origin remains unclear, as does the standard model of progression from benign tumour to malignant carcinoma.

Discord with the progression model is exemplified in the skin, which carries a high mutation burden[6]. Asymptomatic normal skin carries frequent mutations in *TP53* (refs 7,8) and *NOTCH*[6,8]. Classic chemical carcinogenesis 7,12-Dimethylbenz[a]anthracene (DMBA)/12-O-tetradecanoylphorbol-13-acetate (TPA) experiments demonstrate *Hras* mutations can lie dormant in the skin (without the addition of TPA), at no obvious consequence to the tissue[9]. Indeed even when *Ras* mutation is targeted to stem cell compartments (for example, LRIG1$^{+ve}$ cells or bulge stem cells[10,11]), this does not lead to cancer unless there is a disruption of tissue homeostasis through wounding. These findings support the hypothesis that homeostasis within stem cell compartments plays an important tumour suppressive role in highly organized structures such as the skin.

We reasoned that, in the absence of wounding, mutations in other oncogenic/tumour suppressor genes might facilitate rapid skin tumorigenesis. Using targeted sequence analysis and whole-exome sequencing (WES), we identify frequent mutation in both transforming growth factor-β (TGFβ) type 1 receptor (*TGFBR1*) and TGFβ type 2 receptor (*TGFBR2*) genes in human primary cutaneous squamous cell carcinoma (cSCC) samples. IntOgen mutation analysis reveals TGFβ signalling as a pathway significantly altered by mutation and functional analysis of several TGFβ receptor mutants indicates that many of these mutations result in loss of function. Pathway activation studies reveal highly localized TGFβ signalling in both normal human and mouse hair follicle bulge stem cells. In murine skin, targeted activation of the RAS/RAF/mitogen-activated protein kinase (MAPK) pathway, coupled with deletion of *Tgfbr1* in LGR5$^{+ve}$ stem cells, promotes rapid development of cSCC, which, in the absence of wounding, may mimic the kinetics of tumour induction in vemurafenib-induced cSCC. Combined *Tp53* mutation/inactivation coupled with *Tgfbr1* loss in LGR5$^{+ve}$ stem cells also results in cSCC with longer latency, providing a model for cSCC development without RAS activation.

## Results

**TGFBR1 and TGFBR2 are frequently mutated in human cSCC**. Cutaneous squamo-proliferative lesions (including keratoacanthomas and cSCC) arise in a significant proportion of patients treated with the type I RAF inhibitor vemurafenib. Such lesions develop within a few weeks of treatment[12,13]. Targeted sequencing has revealed that these lesions contain a high frequency of activating mutations in *HRAS*[6,12,13]. Cutaneous lesions isolated from patients treated with sorafenib (the 'pan-RAF' inhibitor) also harbour mutations in *HRAS*, *TP53* and *TGFBR1* (ref. 14). Employing targeted deep sequencing of 39 squamo-proliferative lesions from seven patients (including cSCC and actinic keratosis (Supplementary Table 1) treated with vemurafenib (using a percentage variance criterion of >10%), we

identified frequent coding mutations in both *TGFBR1* (8/39, 21% of samples) and *TGFBR2* (5/39, 13% of samples), revealing mutation of TGFβ receptors in 28% of lesions (Fig. 1a and Supplementary Data 1). These mutational events were only surpassed in frequency by mutations in *NOTCH1*/*NOTCH2* (56%) and activating mutations of *HRAS* (38%). *TP53* mutations arose in 26% of lesions[6] (Fig. 1a and Supplementary Data 2). In contrast to *NOTCH* (using our mutational call cutoff, see Methods), we did not detect any mutations in TGFβ receptors or *HRAS* in the normal or perilesional skin samples ($n = 6$ from 4 patients, 3 of which had lesions containing TGFβ receptor mutations). These findings imply that a combination of potential mutational inactivation of TGFβ signalling and activation of *HRAS* may be important driving events in vemurafenib-induced skin lesions and skin tumorigenesis.

We next sought to investigate whether loss of TGFβ signalling is a frequent event in sporadic cSCC. We employed targeted 454 pyrosequencing of *TGFBR1* and *TGFBR2* in 91 human primary cSCC samples (Supplementary Table 2) and 21 human cell lines derived from primary cSCC[15], all of which were recently sequenced for common genetic alterations[6]. Using a percentage variance criterion of >10%, we detected mutations of *TGFBR1* in 22% and *TGFBR2* in 30% of primary cSCC samples and 14% of cell lines (Fig. 1b,c and Supplementary Data 3). Overall, mutation of TGFβ receptors occurred in 43% of primary cSCC samples. These mutational events were only surpassed in frequency by mutations in *NOTCH1/2* (86%) and this time *TP53* (63%) (Fig. 1b, Supplementary Data 4 and ref. 6). In sporadic cSCC, oncogenic activation of RAS only occurred in 9% of samples (Fig. 1b, Supplementary Data 4 and ref. 6). We then sequenced normal blood samples from eight patients with sporadic cSCC, whose lesions harboured mutations in TGFβ receptors (Supplementary Data 3) and found no TGFβ receptor mutations. Next, we prospectively collected a further Dundee cohort of seven primary cSCC samples with complementary matched normal distant and perilesional skin (Supplementary Table 3). This cohort demonstrated a comparable spectrum of mutation in our selected gene panel and in both TGFβ receptors (Fig. 1d and Supplementary Data 5). TGFβ receptor mutations were again not identified in either distant or perilesional skin. To assess the potential lesion-specific, non-germline significance of TGFβ receptor mutations, we interrogated the pyrosequencing analysis in depth from all of the samples containing normal matched tissue (Supplementary Data 6). We observed only 8 variant reads out of 1,348 reads in total, in 4 out of 25 matched normal sample reads. Three of these samples were from perilesional skin and probably reflect rare contaminating tumour cells. In comparison, we observed 237 variant reads out of 1,340 reads in the tumour samples. Employing Fisher's exact two-sided tests to compare variant allele frequencies (VAFs) in matched samples, we determined that 17/25 of the TGFβ receptor mutations reached tumour-specific VAF statistical significance, confirming the lesion-specific, non-germline nature of these mutations (Supplementary Data 6).

**TGFBR1 and TGFBR2 mutations are driver events in human cSCC**. Next, we examined a further cohort of 30 primary cSCC samples with matched normal tissue (Supplementary Table 4) employing next-generation WES (see Methods) and interrogated in detail *NOTCH1*, *NOTCH2*, *TP53*, *CDKN2A*, *HRAS*, *KRAS*, *NRAS*, *TGFBR1* and *TGFBR2* genes for mutational and copy number changes (Fig. 2a and Supplementary Data 7). We observed alterations in all of these genes with a similar frequency to that of our previous 454 pyrosequencing analysis. None of the mutational events were found in the matched normal samples

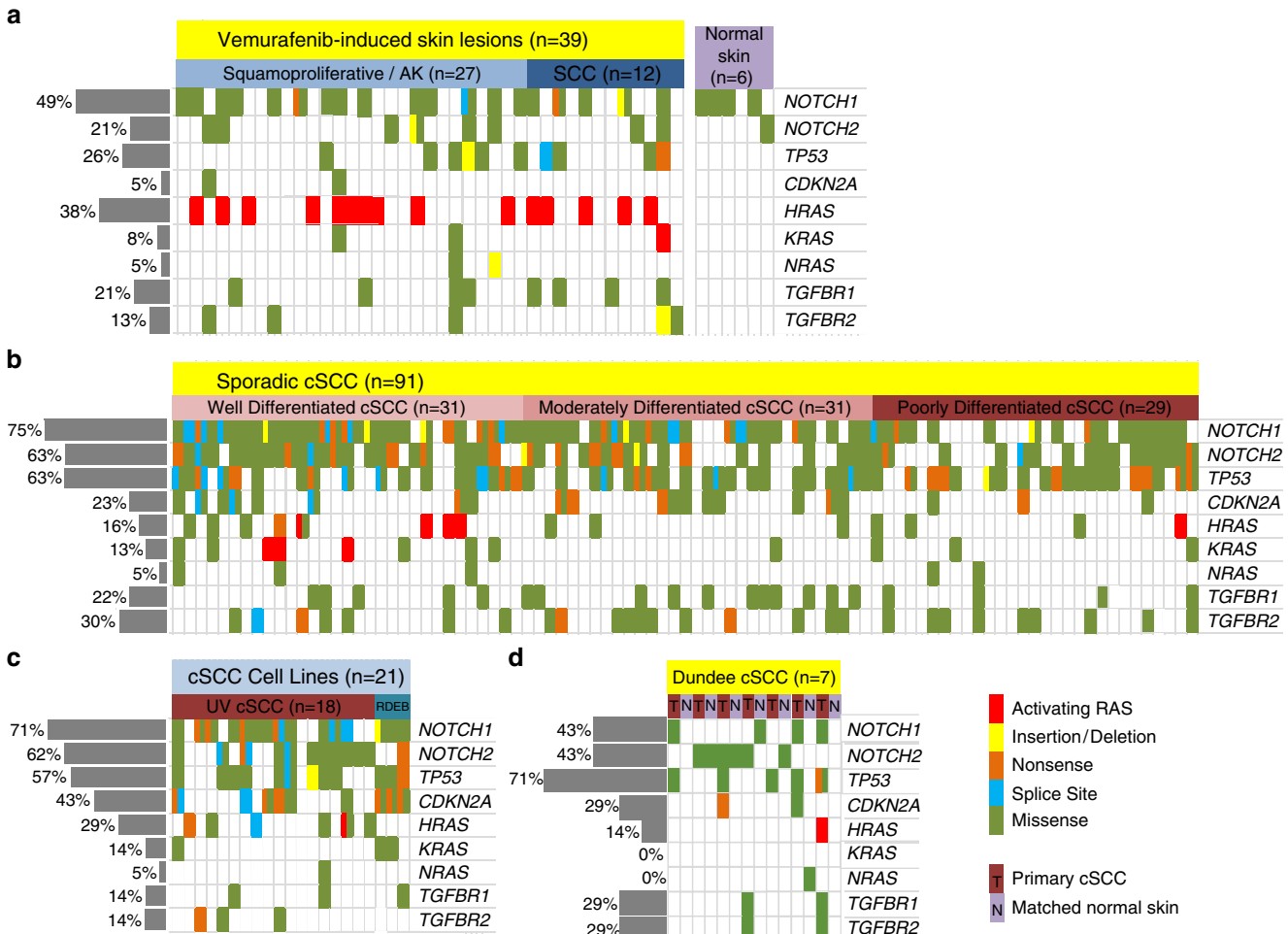

**Figure 1 | TGFβ receptors are frequently mutated in vemurafenib-induced skin lesions and sporadic cSCC tumours.** Mutation frequency, distribution and relationship with pathological features of skin lesions isolated from vemurafenib-treated patients ($n = 39$) (**a**), sporadic cSCC ($n = 91$) (**b**), 21 cSCC cell lines (RDEB, recessive dystrophic epidermolysis bullosa) (**c**); (**a–c**, adapted from ref. 6); 7 sporadic cSCC tumours (T) and normal distant/perilesional skin samples (N) collected in Dundee (**d**). Mutation status for nine genes is indicated and the overall percentage mutation is shown on the left. Each column represents a single case. Colours correspond to specific mutations as shown. Details of clinical parameters are included in Supplementary Tables 1–3 and mutations are included in Supplementary Data 1–5.

and all except two of these were statistically significant (Fisher's exact $t$-test) (Supplementary Data 8). Importantly, we observed changes in *TGFBR1* in 30% of the samples and changes in *TGFBR2* in 40% of the samples with a combined alteration in 53% of samples, confirming a frequent alteration of TGFβ receptor genes in cSCC. Copy number analysis also revealed that loss of heterozygosity occurred in both *TGFBR1* and *TGFBR2* genes including in tumours with missense mutations in *TGFBR2* (Fig. 2a and Supplementary Data 8). Somatic single nucleotide variants (SNVs) of TGFβ receptors were detected in 30% of our samples consistent with our 454 pyrosequencing analysis and the recent sequencing analyses of two North American cSCC cohorts, which, when combined, detected TGFβ receptor proteins altering SNVs in 15.7% of samples[16,17]. Given the high mutational burden of cSCC, it is probable that many mutations identified will be passenger mutations with no functional consequence for tumorigenesis. We investigated the potential functional consequence of the mutations detected by WES employing MutsigCV[18] and IntOgen analysis[19]. MutsigCV detected *TP53*, *CDKN2A*, *NOTCH1* and *NOTCH2* as significant drivers but no *RAS* genes and IntOgen detected *TP53*, *CDKN2A NOTCH1* and *HRAS* as significant drivers but did not identify *NOTCH2*, *KRAS* or *NRAS*

(Supplementary Data 9). Neither analysis detected *TGFBR1* or *TGFBR2* individually as significant drivers (Supplementary Data 9); however, IntOgen pathway analysis revealed TGFβ signalling as a significantly altered signalling pathway (Oncodrive-fm functional impact bias, functional mutation bias[19], $P = 0.0019$; Supplementary Data 10). We assessed the clonality of our candidate driver genes using the ABSOLUTE algorithm[20]. WES data were of sufficient quality for 24/30 exomes and ABSOLUTE analysis revealed purity and ploidy estimates ranging from 0.2 to 0.73 and from 1.78 to 5.79, respectively (Supplementary Data 11). ABSOLUTE clonality analysis indicated that all *NOTCH1*, *CDKN2A* and *RAS* mutations were clonal as were all bar one *TP53*, three *NOTCH2* and one *TGFBR1* mutation, which were subclonal (Fig. 2b,c and Supplementary Data 12). Mutations present in nearly all tumour cells (clonal) would suggest early events and therefore represent initiating 'driver' genes as appears to be the case here for *NOTCH1*, *NOTCH2*, *CDKN2A*, *HRAS*, *KRAS*, *TP53* and, importantly, both *TGFBR1* and *TGBFR2*.

Having established the probable driver event of mutation of *TGFBR1* and *TGFBR2* in our WES data set, we extended this analysis to include our samples assessed by targeted sequencing.

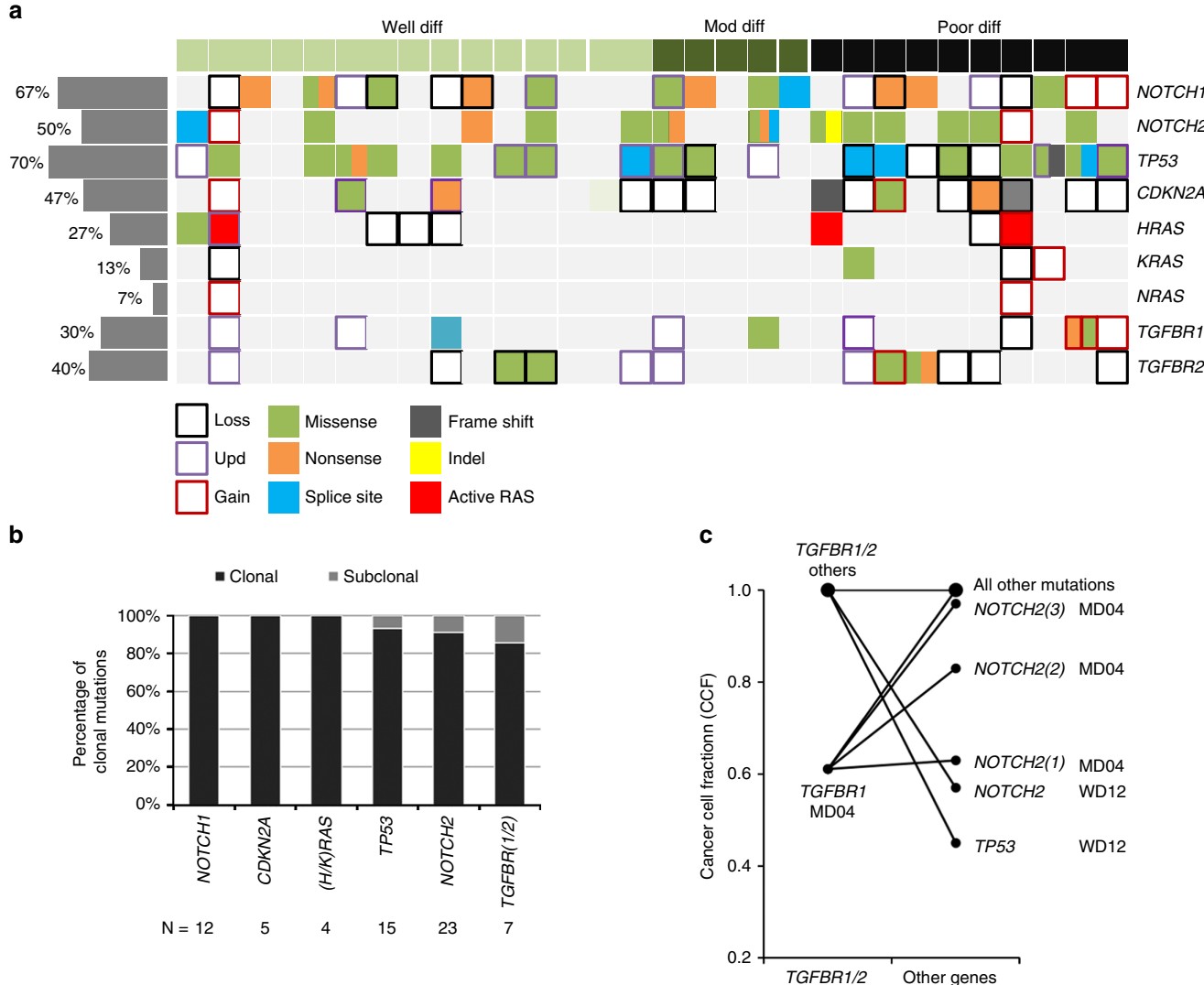

**Figure 2 | Target gene mutation and copy number and clonality analysis in 30 cSCC primary tumours analysed by WES. (a)** Mutation frequency, distribution and relationship with pathological features from 30 cSCC primary tumours. Mutation and copy number status (gain, loss and acquired uniparental disomy) for nine genes is indicated and the overall percentage alteration is shown on the left. Each column represents a single case. Colours correspond to specific mutations and copy number changes as shown. Split columns indicate where more than one mutation type is present in a single case. Details of clinical parameters and mutations are included in Supplementary Table 4 and Supplementary Data 7. **(b)** ABSOLUTE clonality analysis of potential driver genes of cSCC indicates that all nine genes are frequently clonal. **(c)** Cancer cell fraction clonality analysis indicates clonal and subclonal mutations in the indicated tumours.

We first calculated average percentage VAFs for our candidate drivers and these ranged from 48.7% for *CDKN2A* to 20% for *TGFBR1* (Fig. 3a). *TGFBR1* VAF was significantly lower than that of *CDKN2A*, *TP53*, *HRAS*, *NOTCH1*, *TGFBR2* and *NOTCH2* but not *KRAS* and *NRAS* (Fig. 3a and Supplementary Data 13). *TGFBR2* VAF was only statistically significantly lower than *CDKN2A* and *TP53* but equivalent to *KRAS*, *NOTCH2* and *NRAS* (Fig. 3a and Supplementary Data 13). The VAFs of the TGFβ receptors are of a similar range to those observed in other cSCC driver genes. Ultraviolet light is the major oncogenic stimulus of cSCC and the % of mutations conforming to an ultraviolet signature (C-T or G-A transitions) of our candidate drivers ranged from 79.7% in *CDKN2A* to 30.4% in *HRAS* (Fig. 3b), with mutations in both TGFβ receptor genes lying within this range. VAFs were statistically significantly higher for ultraviolet signature mutations for *NOTCH2*, *CDKN2A* and *TGFBR2* (Supplementary Fig. 1a and Supplementary Data 14). If these candidate genes

represent potential driver genes then the mutational consequence should be predicted to change protein function. We classified these mutations as potentially damaging if they were predicted to be so by at least two of the four mutation function prediction programmes SIFT[21], PolyPhen-2 (ref. 22), Provean[23] and Mutation Assessor[24] or were a splice site or PTC mutation (Fig. 3c). Damaging mutation rates ranged from 89% for *TP53* to 53.5% for *TGFBR1* (Fig. 3c and Supplementary Data 15–23) were statistically significantly higher for those with an ultraviolet signature for *NOTCH2*, *TGFBR2* and *TP53* (Fig. 3d and Supplementary Data 24), and damaging mutations had higher VAFs for *NOTCH2*, *CDKN2A*, *TGFBR2* and *NOTCH1* (Supplementary Fig. 1b and Supplementary Data 25). Together, our data suggest that ∼70% of *TGFBR2* and 50% of *TGFBR1* mutations will alter protein function with the potential to drive cSCC development. In its entirety, our analysis conservatively estimates functionally relevant *TGFBR1* and *TGFBR2* mutations in ∼10%

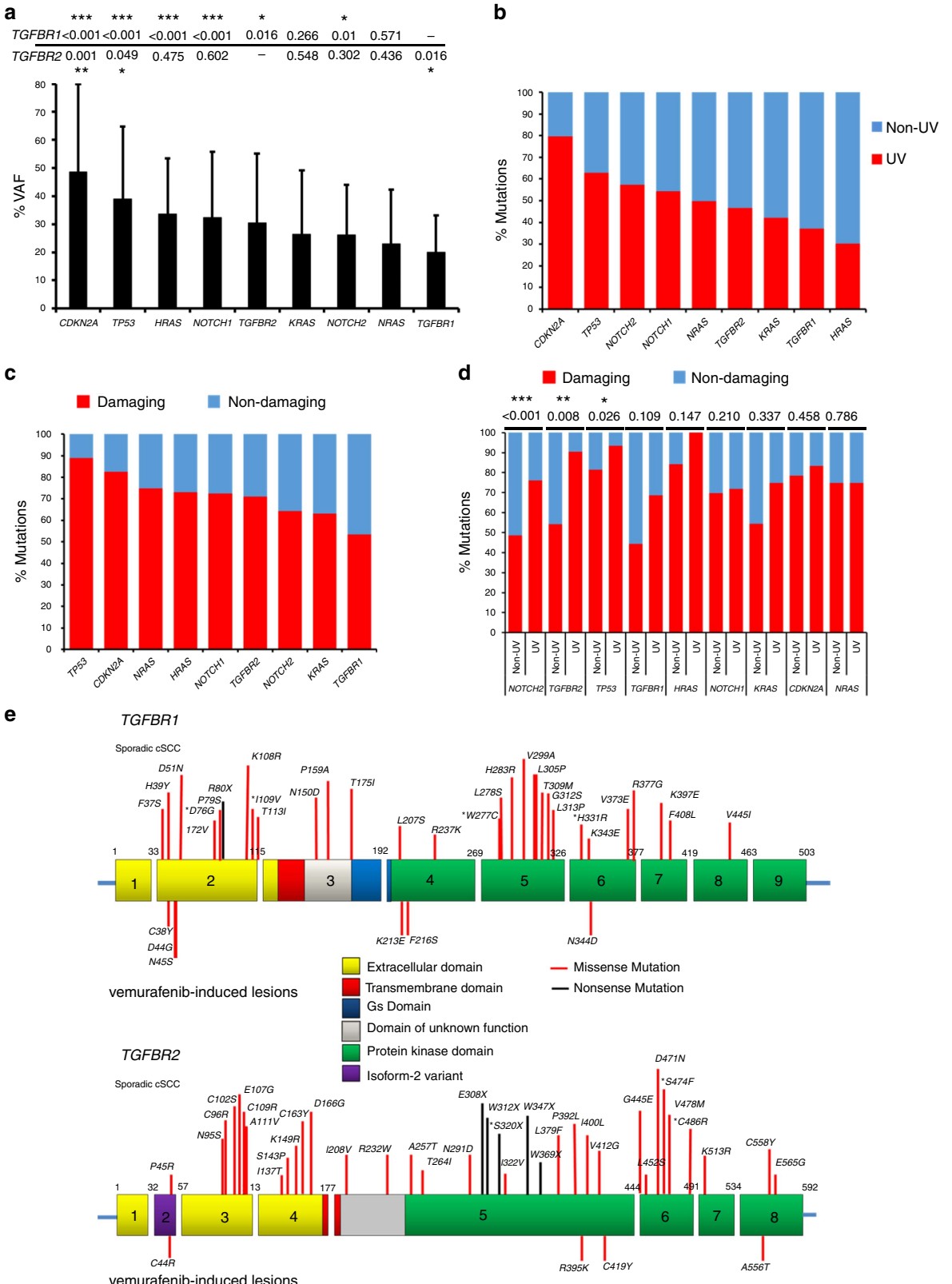

**Figure 3 | Mutational frequencies and spectrum of driver mutations.** Ranked driver gene mutations by (**a**) % VAF (P-values represent Student's t-test (two-tailed) are shown above the figure, (**b**) % ultraviolet spectrum (**c**) predicted mutational consequence of Damaging/Non-damaging and (**d**) combined ultraviolet and damaging analysis (P-values are shown above the figure and represent $\chi^2$ Fisher's exact-test). For all statistics, * defines statistical significance (***P < 0.001, **P < 0.01 and *P < 0.05). Numbers of samples are contained in Supplementary Data 13. (**e**) Domain structures of TGFBR1 and TGFBR2 are shown. Exons are numbered and functional domains colour coded (see key). SNPs identified in TGFBR1 and TGFBR2 are labelled in AA sequence (using UniProtKB codes: P-36897-1, TGFBR1, HUMAN and P37173, P37173-2 and TGFBR2_HUMAN) for sporadic cSCC (above) and Vemurafenib-associated cSCC (below). Asterisked SNPs are those found in cSCC cell lines. Amino acid numbers for TGFBR2 refer to isoform 2.

and ∼16% of samples, respectively, and therefore 20% of cSCC samples could harbour damaging TGFβ receptor mutations.

**TGFβ receptor mutation inactivates canonical Smad signalling.** Identified missense and nonsense mutations were found throughout the coding exons of both *TGFBR1* and *TGFBR2*, occurring in the extracellular and kinase domains of each protein (Fig. 3e). Structural analysis of the extracellular domains of TGFBR1 (Supplementary Fig. 2) and TGFBR2 (Supplementary Fig. 3) indicated mutations occur in, or in close proximity to, highly conserved disulphide bonds, ligand interaction motifs and/ or receptor interaction motifs. These findings suggest significant potential for loss of function[25,26].

TGFβ signals via activation of a heterotetrameric complex of TGFBR2:TGFBR1, resulting in TGFBR1-kinase driven carboxy-terminal phosphorylation of SMAD2 and SMAD3 (ref. 27). Once phosphorylated (PO$_4$), SMAD2 and SMAD3 form hetero-oligomeric complexes with the co-Smad SMAD4 accumulate in the nucleus and regulate gene expression of hundreds of target genes[28,29]. Activity of SMAD-dependent reporter gene constructs and steady-state levels of SMAD2/3 C-terminal phosphorylation can be used as measures of canonical TGFβ signalling. To assess the functional consequence of these TGFβ receptor mutations, we generated a panel of four TGFBR1 and five TGFBR2 mutant expression plasmids from mutations identified in our original targeted sequencing series. We assayed each mutant receptor for functional activity in transient transfection reporter gene assays. TGFBR1 expression plasmids were co-transfected into TGFBR1-null mouse embryonic fibroblasts[30] and TGFBR2 expression plasmids were co-transfected into TGFBR2-null T47D breast cancer cells, in addition to the TGFβ-responsive reporter construct *SMAD7-Promoter Luciferase*[31] (Fig. 4a and b, respectively). Wild-type TGFβ receptor expression elevated reporter activity over empty vector controls, which was further elevated by TGFβ treatment (Fig. 4a,b). We confirmed this activity was dependent on intact SMAD binding elements in the *SMAD7* promoter (Supplementary Fig. 4a,b). The *TGFBR1* mutants H331R and W277C, and all of the *TGFBR2* mutants (S474F, C486R, C96R, R2323W and A556T), failed to efficiently activate the reporter gene, despite similar levels of expression of the receptors, as assayed by western blotting (Fig. 4a,b). These findings indicate that mutation of *TGFBR1* and *TGFBR2* in cSCC frequently results in a loss of ability to activate canonical SMAD signalling. To demonstrate corollary of these findings in primary human tissue, we then established conditions to monitor C-terminal PO$_4$-SMAD3 levels using a C-terminal Ser433/ Ser435 PO$_4$-SMAD3-specific antibody in cSCC by immunohistochemistry (IHC) (Supplementary Fig. 5). We measured PO$_4$-SMAD3 activity in eight primary tumours harbouring wild-type receptors and eight primary tumours harbouring mutant TGFβ receptors with a combined VAF of >20% (Supplementary Data 26). Wild-type tumours exhibited readily detectable PO$_4$-SMAD3 activity, whereas mutant tumours showed significantly reduced PO$_4$-SMAD3 activity (Fig. 4c and Supplementary Fig. 6), consistent with our observation that mutation of TGFβ receptors results in loss of canonical SMAD signalling activity. Both wild-type and mutant tumours exhibited heterogeneity of staining, consistent with our previous observations that cSCC is heterogenous in nature[6] and with the VAFs observed in mutant tumours.

Finally, we used primary human cSCC cell lines to assess whether TGFβ receptor mutation results in a loss of TGFβ signalling. Exogenous treatment of normal human keratinocytes (NHKs) with TGFβ1 resulted in a dose-dependent decrease in cell proliferation (Fig. 4d). The *TGFBR2* mutant harbouring cell lines

SCCIC8 and SCCIC12 (Supplementary Data 3) failed to respond to exogenous TGFβ stimulation by either PO$_4$-SMAD activation (Supplementary Fig. 7) or by any effect on cell proliferation (Fig. 4d). Co-transfecting these *TGFBR2* mutant cells with either empty vector, or wild-type TGFBR2 expression plasmids in addition to a green fluorescent protein (GFP) expression plasmid, we measured cell proliferation in real time using Incucyte-Zoom imager over 6 days. Cell proliferation of the GFP$^{+ve}$ cells indicated that cells expressing wild-type TGFBR2 proliferated at a slower rate in the presence of exogenous TGFβ (Fig. 4e). The degree of inhibition was commensurate to the degree of restoration of SMAD activity as measured using the multimerized SMAD binding element reporter gene *CAGA$_{12}$-Luciferase*[32] (Supplementary Fig. 7c,d). These findings indicate that re-expression of wild-type TGFBR2 restores canonical TGFβ signalling and proliferative inhibition, confirming mutational loss of TGFβ tumour suppressive activity.

**Matrix cells exhibit active TGFβ signalling.** Given this potential aetiological loss of TGFβ signalling, we sought to identify sites of active TGFβ signalling in normal skin, to gain insight into the cellular origin of cSCC RAF inhibitor-induced lesions. PO$_4$-SMAD3 activity was barely detectable by IHC analysis in normal human epidermis (Supplementary Fig. 8) but showed strong immunoreactivity in the hair matrix zone of anagen hair follicles (Fig. 5a and Supplementary Fig. 8). PO$_4$-SMAD3 positivity was also detected in the hair matrix of anagen hair follicles in mouse back skin (Fig. 5a). In anagen, the hair follicle transit-amplifying (TA) cells are localized in the matrix and are positive for Sonic hedgehog (SHH)[11]. Elegant studies by Blanpain and colleagues[11] have demonstrated these cells are unable to act as a cell of origin for papilloma formation, even when both oncogenic *Kras* and *Tp53* were targeted. This suggests that these PO$_4$-SMAD3$^{+ve}$ hair matrix cells are unlikely to be the cell of origin for the rapid cSCC observed in humans following RAF inhibitor treatment. To investigate this in the mouse, we tested whether *Tgfbr1* deletion could permit the transformation of TA cells. RAF inhibitors stimulate paradoxical activation of the MAPK pathway in cells with wild-type BRAF harbouring upstream pathway activation, via mechanisms such as the following: upregulated receptor tyrosine kinases, oncogenic RAS via RAF dimer formation[33–35] or relief of inhibitory auto-phosphorylation[36]. Circumventing pharmacological enhance-ment of MAPK signalling in the presence of mutated *RAS*, we modelled hyperactivation of the MAPK pathway in the SHH$^{+ve}$ compartment by targeting downstream oncogenic *Braf$^{V600E}$* and oncogenic activation of *Kras$^{G12D}$*. We crossed our previously described *LSL-Braf$^{V600E}$* mice[37], which allow inducible expression of *Braf$^{V600E}$* from the endogenous *Braf* gene, with the *ShhCRE$^{ER}$* strain[38]. This permits tamoxifen-inducible activation of the Cre recombinase in SHH$^{+ve}$ cells. To assess the role of TGFβ signalling in the SHH$^{+ve}$ cells, we then crossed these animals with *Tgfbr1$^{fl}$* mice[30] (Supplementary Fig. 9a,b). No tumours formed in the skin of *ShhCRE$^{ER}$ Braf$^{V600E}$* and *ShhCRE$^{ER}$ Braf$^{V600E}$ Tgfbr1$^{fl/+}$* mice (Fig. 5b and Supplementary Fig. 10a,b). A small percentage of *ShhCRE$^{ER}$ Braf$^{V600E}$ Tgfbr1$^{fl/fl}$* mice developed minimally proliferative papillomatous lesions (as evidenced by low level 5-bromodeoxyuridine (BrdU) staining) mainly in the lips, but only at long latency (Fig. 5b and Supplementary Fig. 10c). No mice developed cSCC. Mice failed to develop any skin lesions following oncogenic activation of *Kras$^{G12D}$* with or without deletion of *Tgfbr1* in this cell compartment (Fig. 5c). Together, these studies indicate that the SHH$^{+ve}$ cells are unlikely to be the cell of origin for either rapid onset vemurafenib-induced cSCC or sporadic cSCC.

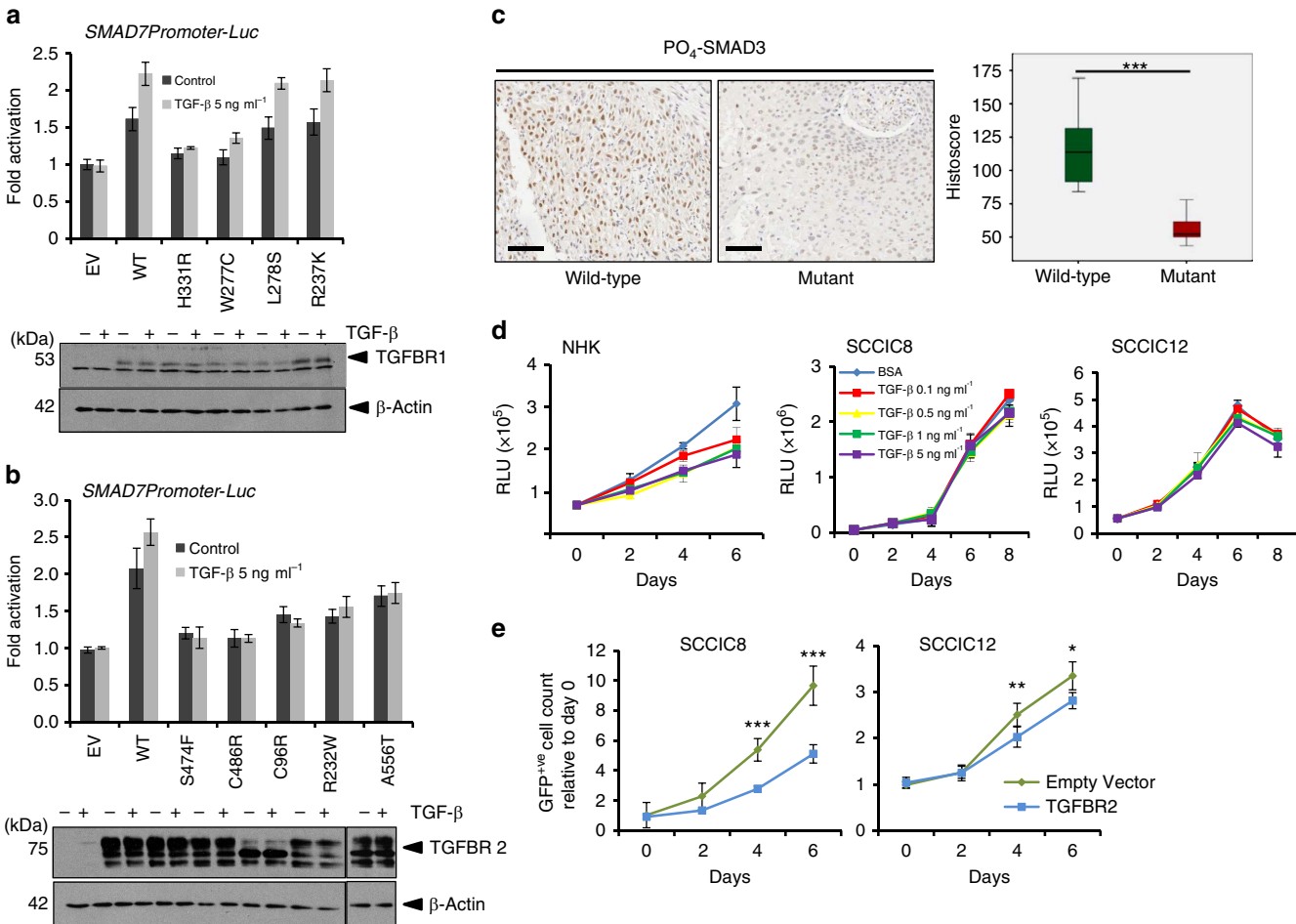

**Figure 4 | Mutation of TGFβ receptors results in loss of function. (a)** Indicated TGFBR1 plasmids were co-transfected into TGFBR1-null mouse embryonic fibroblasts and assayed for *SMAD7-Promoter Luciferase* (*SMAD7 Promoter-Luc*) reporter gene activity and receptor expression levels by western blotting (lower panels) with and without TGFβ stimulation for 4 h. β-Actin is used as a loading control. EV, empty vector control; WT, wild type. Data are mean ± s.d., $n = 3$. **(b)** Indicated TGFBR2 plasmids were co-transfected into TGFBR2-null T47D cells and assayed for *SMAD7-Promoter Luciferase* (*SMAD7 Promoter-Luc*) reporter gene activity and receptor expression levels by western blotting (lower panels) with and without TGFβ stimulation for 4 h. β-Actin is used as a loading control. EV, empty vector control; WT, wild type. Data are mean ± s.d., $n = 3$. **(c)** PO₄-SMAD3 activity was assessed by IHC in wild-type and mutant tumours ($n = 8$, ***$P = 0.001$, Mann–Whitney $U$-test). Representative images are shown. Scale bar, 100 μM. **(d)** Effects of TGFβ stimulation on growth of NHK and cSCC cell lines. Data represent Cell Titre Glo measurement of cell proliferation over the indicated time course of cells treated with the indicated dose of TGFβ1. NHKs and cell lines harbouring mutant TGFBR2 (SCCIC8 and SCCIC12) are shown. Data represent the mean ± s.d. $n = 6$. **(e)** Restoration of wild-type TGFBR2 restores growth inhibition. SCC1C8 and SCCIC12 cells were co-transfected with empty vector control (EV) or wild-type TGFBR2 expression plasmids (TGFBR2) and a GFP expression plasmid. Proliferation of GFP⁺ᵛᵉ cells was assessed using real-time Incucyte Zoom imaging over 6 days. Data represent the mean ± s.d. $n = 6$. *$P < 0.05$, **$P < 0.01$ and ***$P < 0.001$ (Student's $t$-test).

**TGFβ signalling is active in telogen bulge stem cells.** Approximately 90% of human hair follicles are present in the anagen phase of the hair cycle with the remaining 10% existing in catagen or the resting telogen phase. Analysis of human telogen hair follicles revealed highly localized PO₄-SMAD3 staining in the bulge stem cells, characterized in part by Keratin 15 staining (Fig. 6a). This pattern was recapitulated in mouse telogen hair follicles (Fig. 6b), characterized by the expression of the stem cell marker LGR5 (ref. 39). To investigate further, we used the *Lgr5-EGFP-Ires-CREERT2* knockin mouse (hereafter termed *Lgr5CRE^ER^*), where the endogenous *Lgr5* promoter controls expression of enhanced GFP (EGFP) and the CREERT2 fusion protein[40]. IHC analysis for GFP revealed a staining pattern similar of that observed for PO₄-SMAD3 (Fig. 6c). Furthermore, co-immunofluorescence revealed LGR5⁺ᵛᵉ cells (stained for EGFP) are highly enriched for both PO₄-SMAD3 and TGFBR1 (Fig. 6d and Supplementary Fig. 11).

Recent studies indicate that the dermal papilla may provide a source of TGFβ2, activating SMAD signalling in overlying hair germ stem cells[41]. We sorted epithelial EGFP-positive LGR5⁺ᵛᵉ stem cells from murine back skin. Quantitative reverse-transcriptase PCR (qRT–PCR) analysis revealed LGR5⁺ᵛᵉ cells express enhanced levels of *Tgfbr1*, *Tgfb1* and *Tgfb3* messenger RNA when compared with LGR5⁻ᵛᵉ cells, with negligible amounts of *Tgfb2* (Fig. 6e). Expression of *Tgfbr2* was readily detected in GFP⁺ᵛᵉ and GFP⁻ᵛᵉ compartments (Fig. 6e). This indicates enriched autocrine TGFβ signalling in the LGR5⁺ᵛᵉ compartment. We observed high levels of the TGFβ target gene *Smad7* (ref. 42) in LGR5⁺ᵛᵉ cells (Fig. 6e). Together, these findings indicate that autocrine TGFβ signalling is highly localized to the LGR5⁺ᵛᵉ hair follicle bulge stem cells in the mouse and the Keratin 15⁺ᵛᵉ hair follicle bulge stem cells in humans, and that this cell compartment may give rise to both vemurafenib-induced and sporadic cSCC.

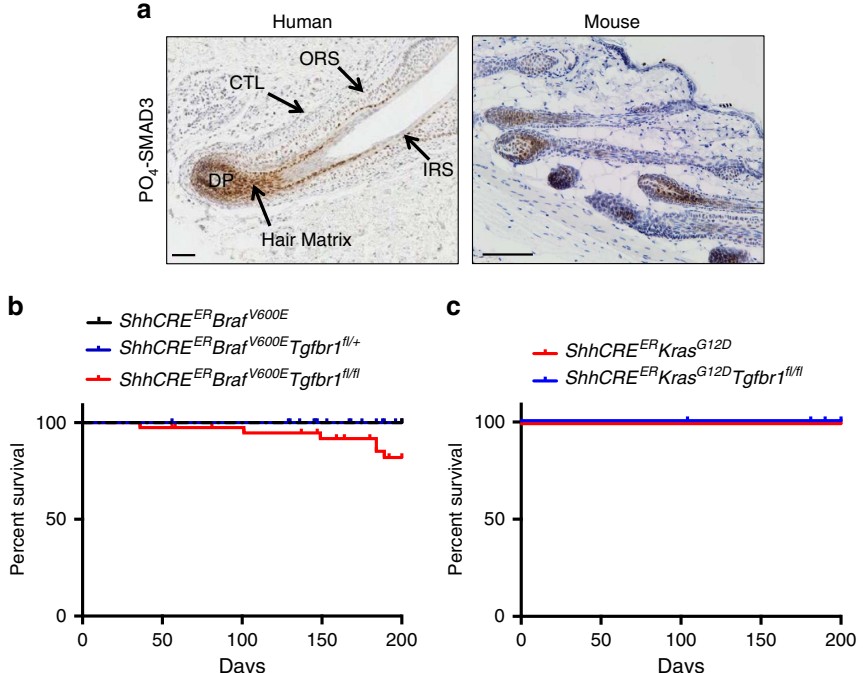

**Figure 5 | TA SHH-positive cells do not allow cSCC development.** (**a**) $PO_4$-SMAD3 IHC in anagen human (left panel) and mouse (right panel) hair follicles reveals immunoreactivity in the hair matrix. CTL, connective tissue layer; DP, dermal papilla; IRS, inner root sheath; ORS, outer root sheath. Scale bar, 100 μm (**b**) Kaplan–Meier survival curve of $ShhCre^{ER}$ $Braf^{V600E}$ ($n=13$), $ShhCre^{ER}$ $Braf^{V600E}$ $Tgfbr1^{fl/+}$ ($n=28$) and $ShhCre^{ER}$ $Braf^{V600E}$ $Tgfbr1^{fl/fl}$ ($n=39$) mice. (**c**) Kaplan–Meier survival curve of $ShhCre^{ER}$ $Kras^{G12D}$ ($n=17$) and $ShhCre^{ER}$ $Kras^{G12D}$ $Tgfbr1^{fl/fl}$ ($n=21$) mice.

**Rapid cSCC formation from Lgr5$^{+ve}$ stem cells**. To investigate the consequence of hyperactivation of the MAPK pathway coupled with ablation of TGFβ signalling in LGR5$^{+ve}$ stem cells, we crossed the $Lgr5CRE^{ER}$ mice with the $LSL$-$Braf^{V600E}$ mice (Supplementary Fig. 9c), or $LSL$-$Kras^{G12D}$ mice and $Tgfbr1^{fl}$ mice (Supplementary Fig. 9d). Loss of TGFβ signalling alone was not sufficient to initiate tumorigenesis (Fig. 7a). Targeted activation of BRAF to LGR5$^{+ve}$ cells resulted in decreased survival, with all mice killed 300 days post induction of the transgene by injection of tamoxifen (median survival 276 days). Although 6 out of 14 mice succumbed to adrenal tumours, 50% of these mice presented with papillomas consistent with LGR5 expression in murine skin (Fig. 7a,c). However, the long latency period suggests $Braf$ mutation requires additional events to facilitate papilloma development. The combined targeted inactivation of one allele of $Tgfbr1$ reduced survival (median survival 231 days) and enhanced both the number of mice with skin lesions and the number of lesions per mouse (Fig. 7a,c and Supplementary Fig. 12a). Inactivation of both $Tgfbr1$ alleles significantly increased the number of tumours per mouse and dramatically shortened both skin tumour-free survival (all mice developing skin lesions within 63 days of induction) and overall survival (median survival 51 days) (Fig. 7a,c and Supplementary Fig. 12a). Phenotypically, these lesions appeared as differentiated papillomas in $Tgfbr1$ wild-type and heterozygous mice (Fig. 7c–e and Supplementary Fig. 12b). Remarkably, in the homozygous $Tgfbr1^{fl/fl}$ mice, tumours appeared as ulcerative cSCC (Fig. 7c and Supplementary Fig. 12b). Elegant work by the laboratories of Blanpain and colleagues[11], and Jensen and colleagues[10] have shown when $Kras$ is targeted to skin stem cells, there is long latency to papilloma formation (similar to the $Braf^{V600E}$ allele described here) and most of these lesions form around areas associated with wounding. Targeted activation of $Kras$ alone mainly failed to produce skin lesions; however, when we targeted inactivation of both alleles of $Tgfbr1$ and the $Kras^{G12D}$ mutation to the LGR5$^{+ve}$

compartment, mice developed rapid cSCC with kinetics comparable to $Braf^{V600E}$ mice (Fig. 7b,c). In addition, Keratin 1 (Fig. 7d) and Keratin 5 staining (Fig. 7e) revealed that cSCC lesions in both the $Braf$ and $Kras$ mice are poorly differentiated cSCC. Importantly, these lesions were highly proliferative (Supplementary Fig. 12c) and never progressed via a papillomatous stage, recapitulating the rapid cSCC onset observed in humans[12,13]. $PO_4$-SMAD3 activity exhibited a dose-dependent reduction in tumours isolated from these mice, indicating loss of TGFβ signalling (Supplementary Fig. 12d,e). qRT–PCR analysis of these tumours revealed loss of $Tgfbr1$ expression (Supplementary Fig. 12f) without any significant change in ligand mRNA expression (Supplementary Fig. 13).

Skin tissue compartmentalization has been recently proposed as a mechanism involved in tissue maintenance[10]. To test whether $Tgfbr1$ deletion perturbed such compartmentalization, we lineage traced LGR5$^{+ve}$ cells by intercrossing $Lgr5Cre^{ER}$ with the $Rosa^{LSL-RFP}$ reporter mice (Supplementary Fig. 9e). We observed that red fluorescent protein (RFP)-positive cells were confined to the hair follicle[39] and were never detected in the sebaceous gland, or interfollicular epidermis regions of $Lgr5Cre^{ER}$ $Braf^{V600E}$ or $Lgr5Cre^{ER}$ $Braf^{V600E}Tgfbr1^{fl/+}$ mice, at early time points post induction (Supplementary Fig. 14). The cSCC arising within $Lgr5Cre^{ER}$ $Braf^{V600E}Tgfbr1^{fl/fl}$ mice were fully recombined and RFP positive. In the normal skin comparator for these tumours, but also at earlier time points, the LGR5$^{+ve}$ cells and their progeny were localized in their normal compartment (Supplementary Fig. 14). These results indicate that perturbation of TGFβ signalling is insufficient to disrupt compartmentalization, but acts as a tumour suppressor in LGR5$^{+ve}$ stem cells.

Given the infrequent coincident activation of $RAS$ genes and mutational inactivation of TGFβ receptors in sporadic cSCC, we finally sought to model this disease by inactivating $Tp53$ function coupled with deletion of $Tgfbr1$ in LGR5$^{+ve}$ cells (Supplementary

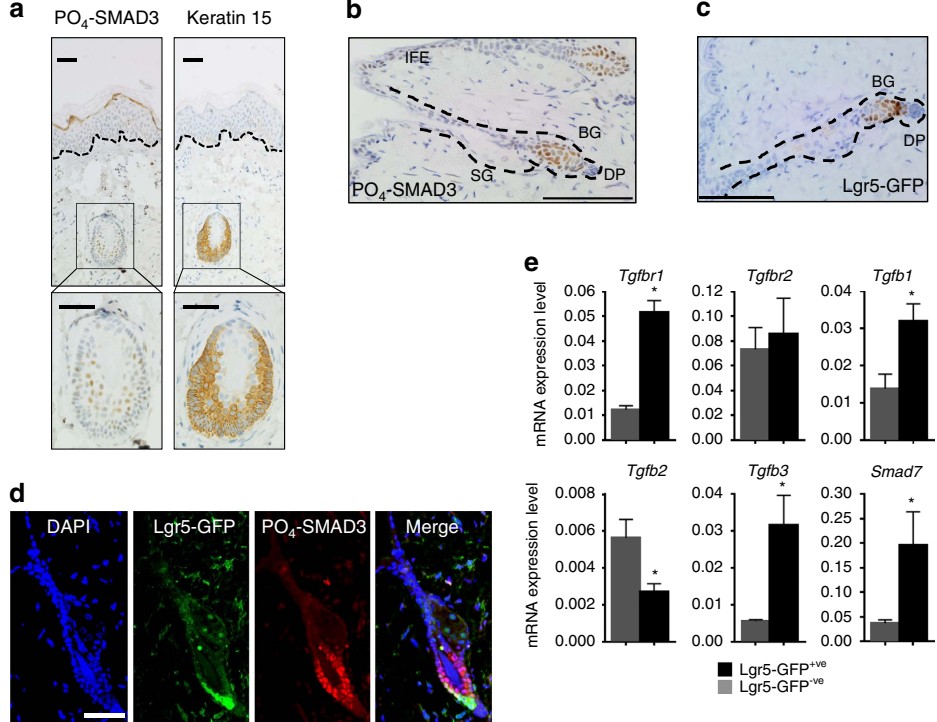

**Figure 6 | TGFβ signalling is active in LGR5 $^{+ve}$ stem cells.** (**a**) IHC analysis of PO$_4$-SMAD3 (left panels) and Keratin 15 (right panels) in human normal skin. Insert shows strong PO$_4$-SMAD3 staining in the telogen hair follicle Keratin 15 $^{+ve}$ bulge stem cells. Scale bar, 100 μm. (**b**) IHC analysis of PO$_4$-SMAD3 in murine normal skin. BG, bulge; DP, dermal papilla; IFE, inter-follicular epidermis; SG, sebaceous gland. (**c**) IHC analysis of LGR5-GFP in murine skin in the telogen phase of the hair cycle. Scale bar, 100 μm. (**d**) Immunofluorescence (IF) analysis of LGR5-GFP and PO$_4$-SMAD3 in murine telogen skin. Nuclei are counterstained with DAPI. Scale bar, 50 μm. (**e**) qRT–PCR analysis of *Tgfbr1*, *Tgfbr2*, *Tgfb1*, *Tgfb2*, *Tgfb3* and *Smad7* in LGR5 $^{+ve}$ ($n = 3$ biological replicates) and LGR5 $^{-ve}$ cells ($n = 3$ biological replicates) freshly isolated from back skin in the telogen phase. Data are shown as ratios to the internal *Gapdh* control with error bars representing s.e.m. Statistical significance *$P = 0.04$ (Mann–Whitney *U*-test, one-tailed test).

Fig. 9f). Knockin of mutant *Tp53* (R172H) coupled with deletion of the wild-type allele had no discernible phenotype (Fig. 8a). Heterozygous knockin or deletion of *Tp53* coupled with homozygous deletion of *Tgfbr1* resulted in the emergence of skin tumours in a few mice (30% and 20%, respectively) with long latency. Combined knockin of mutant *Tp53* with deletion of the wild-type allele of *Tp53* coupled with deletion of *Tgfbr1* resulted in skin tumour development in 81% of mice with increased tumour number at a shorter latency (Fig. 8a,b). These tumours exhibited loss of differentiation expressing low levels of Keratin 1 and higher levels of Keratin 5 (Fig. 8c).

## Discussion

Recent studies have revealed an exceptionally high mutation burden (50 mutations per megabase of DNA[6]) in cSCC[6,16,17]. This rate is second only to that of the commonest skin malignancy basal cell carcinoma[43]. This translates to potentially thousands of mutations per tumour, providing a particular challenge in identifying driver mutations. This challenge is further compounded by varying efficiencies in deep-sequencing technologies and profound tumour heterogeneity[2,6,44,45]. Our studies here reveal that targeted deep sequencing using fluidigm PCR amplification and Roche 454 pyrosequencing can provide a robust platform to identify mutations in *NOTCH1*, *NOTCH2* (ref. 6), *TGFBR1* and *TGFBR2* genes. This approach has also implicated alterations of *NOTCH*, *TP53* and *RAS* in cSCC tumour development[6,46]. We further these studies by revealing mutation of TGFβ receptors in 43% of sporadic human cSCC and 28% of vemurafenib-induced skin lesions (Fig. 1). The prevalent

tumour-initiating event in cSCC is ultraviolet-induced damage, which manifests as C-T and G-A transitions[39]. Approximately 68% of all nucleotide changes observed in our cSCC samples present with this signature[6]. Analysis of mutational signatures in TGFβ receptors reveals that 42% conform to an ultraviolet signature (Fig. 3b and Supplementary Data 13). This figure increases to 56.1% when scored as possibly damaging events via protein function prediction programmes (Fig. 3d and Supplementary Data 24). This indicates that ultraviolet damage may also be responsible for inactivation of TGFβ receptors. Mutation prediction programmes scored 53.5% of *TGFBR1* and 71.1% of *TGFBR2* receptor mutants as damaging, indicating that ~20% of cSCC harbour TGFβ receptor inactivation (Supplementary Data 15 and 16, and Fig. 3c). Subsequent functional analysis of four TGFBR1 mutants and five TGFBR2 mutants indicated that half of the TGFBR1 mutants and all five TGFBR2 mutants were loss of function for canonical Smad signalling, and that tumours harbouring TGFβ receptor mutations had reduced PO$_4$-SMAD3 activity (Fig. 4). Restoration of TGFBR2 expression to TGFBR2-null cell lines restricted cell proliferation (Fig. 4). Taken together, these findings indicate that loss of TGFβ tumour suppressor function is a common event in cSCC.

The assessment of VAFs provides an indication of the clonality of tumours and aids the potential identification of early driver mutations during tumour development[2]. We ranked mutational events by potential order of occurrence in the seven genes we have previsouly implicated in cSCC development and TGFβ receptors by measurement of VAF (Supplementary Data 15 and 16). These analyses indicate that potentially damaging mutations in *TGFBR1* occur early in 25% of tumours harbouring these

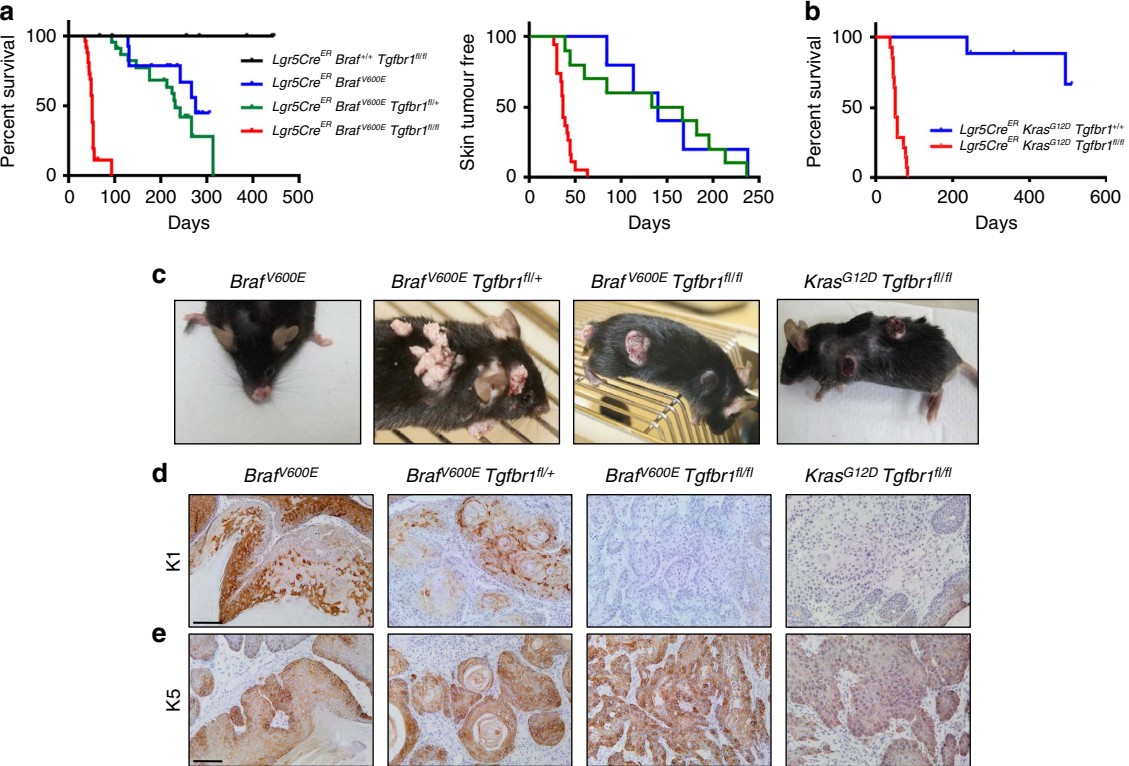

**Figure 7 | Deletion of *Tgfbr1* coupled with BRAF/KRAS activation leads to skin tumorigenesis.** (**a**) Kaplan–Meier survival curve (left panel) of *Lgr5CreER Tgfbr1fl/fl* (n = 12), *Lgr5CreER BrafV600E* (n = 14), *Lgr5CreER BrafV600E Tgfbr1fl/+* (n = 23) and *Lgr5CreER BrafV600E Tgfbr1fl/fl* (n = 26) mice (P ≤ 0.0001 by log-rank, Mantel–Cox). Skin tumour-free survival curve (right panel) of *Lgr5CreER BrafV600E* (n = 5), *Lgr5CreER BrafV600E Tgfbr1fl/+* (n = 10) and *Lgr5CreER BrafV600E Tgfbr1fl/fl* (n = 19) mice. (**b**) Kaplan–Meier survival curve of *Lgr5CreER KrasG12D* (n = 9) and *Lgr5CreER KrasG12D Tgfbr1fl/fl* (n = 14) mice (P < 0.0001 by log-rank, Mantel–Cox). (**c**) Macroscopic pictures of skin tumours from *Lgr5CreER BrafV600E, Lgr5CreER BrafV600E Tgfbr1fl/+, Lgr5CreER BrafV600E Tgfbr1fl/fl* and *Lgr5CreER KrasG12D Tgfbr1fl/fl* mice. (**d**) Representative staining of Keratin 1 (K1) and (**e**) Keratin 5 (K5) on *Lgr5CreER BrafV600E, Lgr5CreER BrafV600E Tgfbr1fl/+, Lgr5CreER BrafV600E Tgfbr1fl/fl* and *Lgr5CreER KrasG12D Tgfbr1fl/fl* mice. Scale bar, 100 μm.

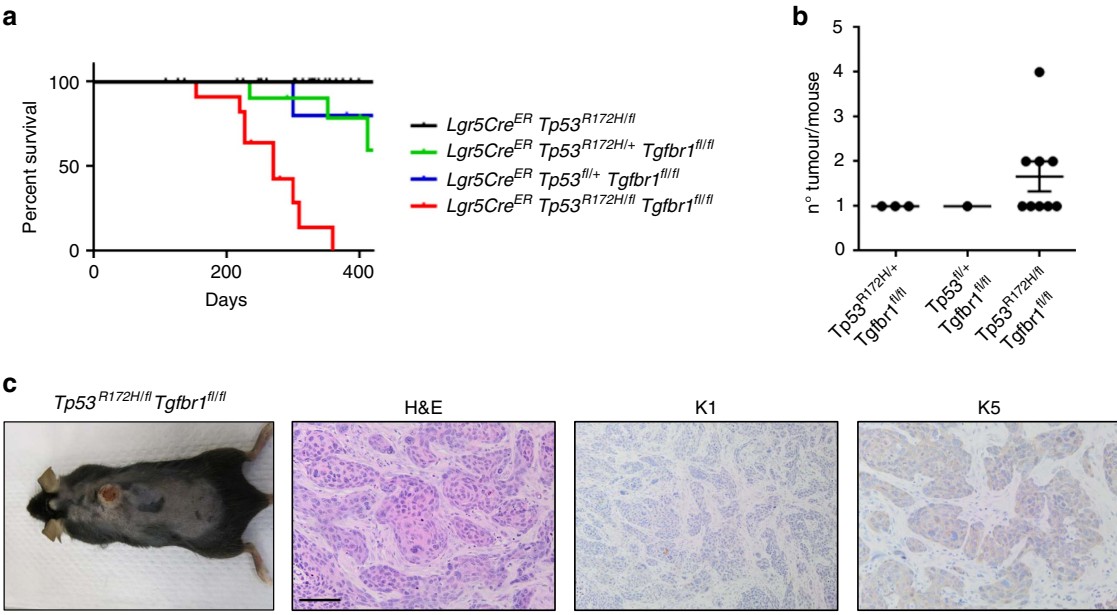

**Figure 8 | Deletion of *Tgfbr1* coupled with *Tp53* mutation/deletion leads to skin tumorigenesis.** (**a**) Kaplan–Meier survival curve of *Lgr5CreER Tp53R172H/fl* (n = 31), *Lgr5CreER Tp53R172/+ Tgfbr1fl/fl* (n = 10), *Lgr5CreER Tp53fl/+ Tgfbr1fl/fl* (n = 5) and *Lgr5CreER Tp53R172H/fl Tgfbr1fl/fl* (n = 11) mice (P ≤ 0.0001 by log-rank, Mantel–Cox). (**b**) Tumour number of indicated genotypes. (**c**) Macroscopic picture of a *Lgr5CreER Tp53R172H/fl Tgfbr1fl/fl* mouse with a skin tumour. Representative staining of haematoxylin and eosin, Keratin 1 (K1) and Keratin 5 (K5) is shown. Scale bar, 100 μm.

mutations and in 42% of tumours harbouring potentially damaging *TGFBR2* mutations. Although this analysis is limited to the 9 genes studied in-depth here (but importantly including *NOTCH* genes previously identified as gatekeeper mutations in cSCC[6]), 11 samples exhibited TGFβ receptor mutations with the highest VAF, indicating that this could be an initiating event in the development of cSCC. Strong support for this hypothesis comes from ABSOLUTE clonality analysis of our WES samples, which revealed that 7/8 TGFβ receptor mutations were clonal and represent probable driver events in these lesions.

Several of our samples display VAFs of 10–20% for the TGFβ receptors, which is not too dissimilar to the VAFs of the other known cSCC tumour suppressors studied here, and probably reflects the heterogeneous nature of cSCC. It is however an intriguing possibility that in some cases low TGFβ receptor VAF may reflect spontaneous regression of TGFβ receptor mutant clones as observed in multiple self-healing squamous epithelioma patients who harbour germline mutations in *TGFBR1* (ref. 47). TGFβ signalling has been demonstrated to play both positive and negative roles in cSCC development in various mouse models[48] acting to limit tumour cell proliferation, and also to promote tumour-initiating capacity and drug resistance[49]. Effects are dependent on the timing of aberrant TGFβ signalling and the cooperating oncogenic driving events (reviewed in refs 48,50–52). We provide evidence that a dose-dependent loss of TGFβ signalling drives tumour progression, emphasizing its role as a major tumour suppressor in the skin. Although the cell of origin in cSCC in humans remains poorly defined[3], our observations indicate that tumours can initiate efficiently and rapidly from LGR5$^{+ve}$ stem cells compared with TA cells, and mutational modulation of two signalling pathways within this cellular compartment is sufficient to drive rapid progression directly to carcinoma, without the need for protracted tumour evolution. The kinetics of this event mimic exactly that observed during development of cutaneous lesions in RAF inhibitor-treated patients[12,13]. Intriguingly this rapid process requires MAPK pathway activation as targeted intereference with *Tp53* function coupled with *Tgfbr1* loss results in the development of skin tumours with long latency. Importantly, our studies revealed highly localized TGFβ signalling in Keratin 15$^{+ve}$ bulge stem cells in human telogen hair follicles, mirrored exquisitely the location of specific autocrine TGFβ signalling activity identified in LGR5$^{+ve}$ bulge stem cells of murine telogen hair follicles. It has been proposed that stem cell quiescence acts as a tumour suppresive mechanism in murine skin, and that LGR5$^{+ve}$ stem cells are refractory to oncogenic transformation[53]. Our data clearly indicate that oncogenic activation of the RAS/RAF/MAPK pathway, or p53 modulation, coupled with loss of TGFβ signalling, is capable of leading to tumour development from this compartment. As we demonstrate that mutational inactivation of TGFβ receptors is a frequent event in human cSCC, and that TGFβ signalling is highly localized to stem cells in normal skin, we propose that these cells represent a cell of origin for human cSCC. It remains possible that loss of TGFβ signalling may also contribute to cSCC development from other cell compartments in the skin and this warrants further investigation.

Our data, both in human and mice, indicate that TGFβ signalling inactivation can be an initiating event in sporadic cSCC. This is clearly the case in multiple self-healing squamous epithelioma where germline loss-of-function mutations in *TGFBR1* have been identified as the underlying genetic lesion[47]. We speculate that activation of the RAS/RAF/MAPK pathway, or p53 modulation, may be a cooperating event in the development of this disease, and that these tumours may originate from the bulge stem cell compartment. TGFβ signalling inactivation may also occur following the acquisition of other driving mutational

events and act as a limiting factor for tumour development. Intriguingly, initial clinical trials targeting systemic TGFβ inhibition with GC1008 (a pan-TGFβ neutralizing antibody) have also reported the occurrence of spontaneous cSCC as a side effect[54]. This provides further compelling support for the tumour suppressive role of TGFβ in skin carcinogenesis.

## Methods

**Samples.** Ethical approval for this investigation was obtained from the East London and City Health Authority and the Tayside Tissue bank local ethics committee, and the study was conducted according to the Declaration of Helsinki Principles. All patients participating in this study were from dermatology and plastic surgery units in the United Kingdom and all provided written, informed consent. Punch biopsies of cSCC tissue were collected and processed as previously reported[6]. NHKs were isolated from normal skin samples according to previously published protocols[55]. Human tumour cell lines SCCIC1, SCCIC4, SCCIC8, SCCIC15, SCCIC12, SCCIC18, SCCIC19, SCCIC21, SCCT1, SCCT2, SCCT6, SCCT8, PM1, MET1, MET4, SCCT9, SCCT10, SCCT11, RDEBSCC2, RDEBSCC3, RDEBSCC4 and NTERT cells were established by our laboratories and were cultured as described[15]. TGFBR1-null MEFs[30] and T47D cells (ATCC) were maintained in DMEM and RPMI medium supplemented with 10% FCS, respectively. Mycoplasma contamination checks were carried out on all cultures as routine and all lines were confirmed mycoplasma negative.

**454 Sequencing.** *TGFBR1* and *TGFBR2* primers were designed and validated by Fluidigm (Fluidigm Corporation, San Francisco, CA) as per recommended guidelines for Roche Titanium sequencing (Roche, Mannheim, Germany). Primers for *NOTCH1*, *NOTCH2*, *TP53*, *CDKN2A*, *HRAS*, *KRAS* and *NRAS* were previously described[6] and all primer sequences are listed in Supplementary Data 27. Each primer included sample-specific Fluidigm 454 barcode primer and adapter sequences. Sequencing was performed in the same manner as our previous study[56]. Briefly, for thermal cycling a Fluidigm FC1 Cycler was used. The libraries were normalized and pooled before purification using Agencourt AMPure XP system (Beckman, UK). Library components were clonally amplified utilising the GSJunior emPCR Lib-A Kit (Roche) by inputting one molecule of library DNA per capture bead. Pyrosequencing was done using the GS Junior system (Roche/454 Life Sciences).

**454 Variant analyses.** Variant analysis was performed as previously described[6]. Briefly, reads were mapped to the hg19 build of the human genome using LASTZ via the public GALAXY instance and filtered to exclude those mapping to <100 loci using tools available through GALAXY.

**Coding variants and splice site detection.** Pileup files were generated and filtered using SAMTools[57]. Variants present in a single read or <10% of the total reads were excluded using a custom java programme available from https://github.com/mattsouth/laszt-variant-filter (last accessed 6 May 2013). Coding variants were called against the RefSeq gene list using the amino acid tool via the public GALAXY instance. Variants present in <3 reads were excluded. Variants present in >1 independent sample and adjacent to a homopolymer >3 bases were excluded, unless present in COSMIC[58]. Variants present in >30 samples were excluded, unless present in COSMIC. All variants present in the exome variant server database (http://evs.gs.washington.edu/EVS/) were excluded, unless present in COSMIC. Splice sites were called from the pileup variant list if present in >4 reads and within 2 bases of Refseq coding sequence using Excel (Microsoft Inc., CA).

**WES data analysis.** Twenty previously published cSCC whole exomes[6] were re-analysed with the addition of ten new cSCC whole exomes with the overall mean coverage of 63× (Supplementary Table 4), using a previous pipeline[59]. SNVs and short indels were identified using the Strelka pipeline[60] with a minimum coverage of ten reads at the targeted sites. Annotation of somatic variants was performed using the Oncotator tool[61]. Mutations in our targeted genes were further identified across the 30 cSCC WES samples.

**Copy number analysis using WES data.** Two independent approaches were applied. First, to generate SNP and indel variant genotyping information, the tumour-normal pair was processed together against the reference genome using the VarScan2 germline variants calling method mpileup2cns[62]. The minimum coverage for identified sites was ten reads for both tumour and normal. Next, the logR and BAF (B-allele frequency) files were created based on the tumour-normal pair genotyping information, with the depth information normalized by dividing the depth of each variant by the median depth across all variants. The ASCAT R packages[63] were then used to perform allele-specific copy number analysis, to identify copy number aberrations and loss-of-heterozygosity regions. The second approach was based on number of reads aligned to each exon between the tumour

and normal pair. VarScan2 copy number calling method was first applied. Raw copy number calls were adjusted as previously reported[64]. Finally, results from the two approaches were cross-compared, to produce the final copy number aberrations and acquired uniparental disomy calls for targeted genes.

**Identification of potential cancer drivers and significantly mutated pathways.** Based on all mutations identified from the 30 cSCC WES data set, we used the IntOGen platform[19] to identify significantly mutated genes and pathways, based on the significance (P-value) of the FM bias (that is, the bias towards the accumulation of mutations with high functional impact). The significantly mutated signalling pathways (based on the IntOGen Oncodrive-fm functional impact bias, FM bias $P < 0.05$) were further selected (Supplementary Data 10). MutsigCV[18] was also used to detect significant genes with point mutations above the background mutation rate.

**Estimating the clonality of mutations.** For the somatic mutations of TGFBR1/2, TP53, CDKN2A, NOTCH1/2 and RAS genes identified by WES (Supplementary Data 7), we further classified them as clonal or subclonal on the basis of the posterior probability that the cancer cell fraction exceeded 0.95 using ABSO-LUTE[20]. Numbers of reads supporting the reference and alternative alleles were extracted and the copy number segmentation files were generated based on the DNAcopy CBS segments using WES data. Mutations with the somatic clonal probability $> 0.5$ were classified as clonal with high confidence. Those mutations with clonal probability $> 0.25$ but with very small subclonal probability scores were also called clonal (Supplementary Data 12). Tumour purity and ploidy were also estimated (Supplementary Data 11). For samples with TGFBR1/2 mutations, cancer cell fractions for TGFBR1/2 were further compared with those for other genes, to determine the clonality orders.

**Functional prediction of mutations.** A combination of four approaches were used to predict the functional impact of identified mutations by targeted sequencing, (i) SIFT[21], which uses sequence homology and protein conservation to predict the effects of all possible substitutions at each position in the protein sequence; (ii) PolyPhen-2 (ref. 22), which predicts possible functional impact of an amino acid substitution on the structure and function level using physical and comparative considerations; (iii) Provean[23], which predicts the damaging effects of SNVs and indels using a versatile alignment-based score; and (iv) Mutation Assessor[24], which measures the functional impact scores for amino acid residue changes using evolutionary conservation patterns derived from aligned families and sub-families of sequence homologues within and between species. Mutations predicted as functional damaging by at least two of the four approaches were classified being potentially damaging/deleterious.

**In vivo analyses.** All experiments were performed under the UK Home Office guidelines. Mice were segregating for C57BL6J and S129 background. Alleles used throughout this study were: Lgr5-cre-ER[T2](ref. 40), ShhCre[ER] (ref. 38), Braf[V600E] (ref. 37), Kras[G12D] (ref. 65), Tgfbr1[fl30], Rosa[LSL-RFP] (ref. 66), Tp53[fl] (ref. 67) and Tp53[R172H] (ref. 68). A mix of males and females were used. Recombination in the Lgr5-cre-ER[T2] mouse model was induced with one intraperitoneal injection of 3 mg Tamoxifen (Sigma) followed by one injection of 2 mg Tamoxifen for 3 days. Mice were induced post 7 weeks of age. Recombination in the ShhCre[ER] mouse model was induced with one intraperitoneal injection of 2.5 mg Tamoxifen. Mice were induced post 28 days of age. For proliferation analysis, mice were injected with 250 μl of BrdU (Amersham Biosciences) 2 h before being sampled.

**FACS analysis.** Epidermis was prepared as previously described[69]. Briefly, fat was scraped from the mouse back and left at 37 °C in a dish (dermis down) in 0.25% of Trypsin/EDTA (Invitrogen, Carlsbad, CA) for 90 min. Epidermis was removed using a scalpel and dissociated by pipetting. Cells were filtered through a 40 μm strainer, centrifuged at 250 g for 5 min and washed with PE (PBS/EDTA). Cells were washed with 0.1% BSA/PE, centrifuged at 250 g for 5 min and used for Lgr5-GFP sorting.

**Immunohistochemistry.** IHC was performed on formalin-fixed skin sections. Standard IHC techniques were used throughout this study. Primary antibodies were as follows: TGFBR1 (Santa Cruz, V22, 1:100), PO4-SMAD3 (Abcam, EP823Y, (52903), 1:50), GFP (Abgent, 168AT1211, 1:100), Keratin 1 (Covance, AF109, 1:1,000), Keratin 5 (Covance, AF138, 1:4,000), Keratin 15 (Abcam, 80522 (LHK15), 1:1,000), KI67 (Thermo, RM-9106-S) and BrdU (BD Biosciences, 347580, 1:200). Mouse PO4-SMAD3 score was performed in a blinded manner. For each antibody, staining was performed on at least three mice of each genotype and at least six sections of normal human skin. Representative images are shown for each staining. PO4-SMAD3 antibody was optimized for IHC use using formalin-fixed paraffin-embedded SCCIC4 cells treated with and without recombinant TGFβ1 or the TGFBR1 kinase inhibitor SB-431542 (ref. 70) (Supplementary Fig. 5). PO4-SMAD3 IHC scoring was performed in a blinded manner using the histoscore method.

**RNA isolation and quantitative PCR.** RNA was isolated using a Qiagen RNeasy Mini Kit (Qiagen, Crawly, West Sussex, UK) according to the manufacturer's instructions. DNA-free kit (Ambion/Applied Biosystems, Warrington, UK) was used to remove genomic DNA contamination according to the manufacturer's instructions. One microgram of RNA was reverse transcribed to complementary DNA using a DyNAmo SYBR Green 2-step quantitative PCR kit (Finnzymes, Espoo, Finland) in a reaction volume of 20 μl. Glyceraldehyde 3-phosphate dehydrogenase (GAPDH) was used to normalize for differences in RNA input.

**qRT–PCR primers.** qRT–PCR primers were as follows. mTgfbr1 F-5′-TGCCATA ACCGCACTGTCA-3′, mTgfbr1 R-5′-AATGAAAGGGCGATCTAGTGATG-3′; mTgfbr2 F-5′-CCGGAAGTTCTAGAATCCAG-3′, mTgfbr2 R-5′-TAATCCTTC ACTTCTCCCAC-3′; mTgfb1 F-5′-AGCCCGAAGCGGACTACTAT-3′, mTgfb1 R-5′-TTCCACATGTTGCTCCACAC-3′; mTgfb2 F-5′-TTTAAGAGGGATCTT GGATGGA-3′, mTgfb2 R-5′-AGAATGGTCAGTGGTTCCAGAT-3′; mTgfb3 F-5′-CGCACAGAGCAGAGAATTGA-3′, mTgfb3 R-5′-GTGACATGGACAG TGGATGC-3′; mSmad7 F-5′-TCAAGAGGCTGTGTTGCTGT-3′, mSmad7 R-5′-TGGGTATCTGGAGTAAGGAGGA-3′; and mGapdh F-5′-GAAGGCCG GGGCCCACTTGA-3′, mGapdh R-5′-CTGGGTGGCAGTGATGGCATGG-3′.

**Western blotting.** Cells were lysed directly in 4 × SDS sample buffer at 60–80% confluence. Lysates were subjected to standard SDS–PAGE. Bands were detected using enhanced chemiluminescence solution (Amersham). Secondary antibodies used throughout were horseradish peroxidase-conjugated polyclonal goat anti-mouse Ig (Dako, P0448, 1:2,000) and horseradish peroxidase-conjugated polyclonal goat anti-rabbit Ig (Dako, P0260, 1:2,000). Primary antibodies were PO4-SMAD3 (Abcam, 52903, 1:1,000), SMAD3 (Cell Signaling, 9523, 1:1,000), TGFBR1 (Santa Cruz, 398 (V22), 1:500) and TGFBR2 (Santa Cruz, 17792, (E6), 1:500). For TGFBR2 western blottings, lysates were prepared directly from transfected cells using the Dual-luciferase cell lysis buffer (Promega, UK). For TGFBR1 western blottings, parallel transfections to the luciferase assays were performed and samples were lysed directly in 4 × SDS sample buffer. Original uncropped western blot scans are also provided (Supplementary Fig. 15).

**Plasmids.** The full-length wild-type human TGFBR1 and pathogenic mutants, amplified with BglII/NotI restriction sites, were shuttled into pCMV5 mammalian cell expression vectors onto the BamHI/NotI sites. The full-length wild-type human TGFBR2 and pathogenic mutants were sub-cloned into pCMV5 using the BamHI/NotI restriction sites. Site-directed mutagenesis was carried out using the QuickChange method (Stratagene) but substituting the Taq with KOD Hot Start DNA polymerase (Novagen). All DNA constructs were verified by DNA sequencing (by the DNA Sequencing Service at University of Dundee; www.dnaseq.co.uk). GFP expression plasmid was from Amaxa.

**Transient transfection analysis.** All transfections were performed in 24-well format in biological triplicate using LipofectAMINE 2000 (Invitrogen) according to the manufacturer's instructions. Cells were transfected overnight with 100 ng of reporter gene (SMAD7-Promoter Luciferase or CAGA12-Luciferase) and 10 ng of internal Renilla luciferase control (pRL-TK, E2241, Promega) with empty vector (pCMV5, 211175, Stratagene), wild-type or mutant TGFβ receptor plasmids (range 150–300 ng). Recombinant human TGFβ1 (Peprotech) was dissolved in 4 mM HCL/1 mg ml$^{-1}$ BSA and used at final concentration of 5 ng ml$^{-1}$, and cells were treated for 4 h before harvest. Luciferase activities were measured using the Dual Luciferase assay (Promega) and firefly luciferase activity was normalized to Renilla luciferase activity.

**Cell proliferation assays.** Cells were seeded at a density of 500–1,000 cells per well of 96-well plates in keratinocyte media (RM$^+$) without growth factors and incubated overnight. Cells were fed 50 μl of medium supplemented with treatment and controls every 2 days until harvest. All cultures were performed in sextuplet ($n = 6$). Cells were assayed for proliferation using the CellTitreGlo Luminescent Cell Viability assay (NHKs) as per the manufacturer's instructions (Promega; luminesence was measured on a Berthold Orion II microplate luminometer) or IncucyteZoom Live cell imager.

**Data availability.** The WES data for the 30 samples have been deposited in the European Genome-phenome Archive under accession code EGAS00001001892. The authors declare that all other relevant data supporting the findings of this study are available within the article and its Supplementary Information files. Additional information can be obtained from the corresponding authors (G.J.I. and O.J.S.).

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

## Acknowledgements

O.S. is supported by a Cancer Research UK core grant (A21139) and an ERC starting grant (311301). P.C. is supported by FP7 Health CP-IP - Large-scale integrating project grant (278568). D.F.V. is supported by ERC Starting grant (311301). A.M.R. is supported by Cancer Research UK centre grant (A12976). L.C.S. was supported by WWCR grant 11-0788. G.J.I. was supported by WWCR fellowship (03-0900). G.J.I., I.M.L., C.M.P., C.A.H., K.J.P., A.M., C.P. and A.P.S. were supported by a Cancer Research UK programme grant (A13044) and an ERC grant (250170). J.W., A.N. and C.C. were supported by a Cancer Research UK centre award to Barts Cancer Institute. We thank the research and scientific services at the CRUK Beatson Institute in general.

## Author contributions

O.J.S., G.J.I., P.C., A.M.R., D.F.V., J.W. and A.P.S. contributed to study design. P.C., A.M.R., D.F.V., S.L. and D.A. contributed to the data acquisition. P.C., A.M.R., D.F.V., J.W., A.N., S.L., R.A.R., D.A., P.J.C.V., A.M., C.P., J.H.S.D., J.L., S.W., L.C.S., G.P.S., K.J.P., C.M.P., C.A.H., I.R.L., H.C., N.B., S.K., C.P., R.M., C.C., A.P.S., O.J.S. and G.J.I. contributed to the data analysis and interpretation of the data. O.J.S., G.J.I., P.C., A.M.R., D.F.V., A.P.S. and J.W. contributed to drafting the manuscript.

## Additional information

**Competing financial interests:** The authors declare no conflict of interest.

