## [Peer Review File · Nature Communications]

Reviewer #5 (Remarks to the Author):

The presented article by Cammareri and colleagues is of high importance as it provides the possible mechanism of the cSCC development upon pharmacological BRAF inhibition.

The stringent statistical analyses as well as the various algorithms used to predict whether the identified mutations are deleterious convincingly demonstrate the TGFBR1 and 2 mutations are driver events in cSCC. Indeed the authors used normal tissue in order to validate the mutations are somatic and the fact the authors used several tumor cohorts is impressive. Furthermore the authors genetic and functional data do support the fact that the driver events they identified are of the tumor suppressor category and fit similar tumor suppressors previously identified in the literature.

The authors not only prove the fact that TGFBR1 and 2 mutations are drivers using genetic tools but also go further to delineate the cell of origin of cSCC to be Lgr5 +compartment in hair follicle bulge. Indeed the leap from TGFBRs activity in the skin compartments and the cell of cSCC origin is clear, but taking the apparent lack of the available system to trace human skin Lgr5+ stem cell lines harboring BRAFV600E mutation and TGFBR inactivation makes it challenging to make any more convincing transition. Importantly, the paper significantly benefit from the comparison of the developed mouse models and human samples as it demonstrated that the mouse model recapitulate histopathology of human phenotype of cSCC.

The authors further investigate the pathways that need to converge for disease progression. The finding that concomitant hyperactivation of the MAPK pathway is needed for RAF-inhibitor induced TGFBR-dependent cSCC formation in Lgr5+ skin stem cells is well-designed and inclusion of TP53 model clearly supports the conclusion.

Minor remark:

Overall the study is well read and the flow of the results presented is easy to understand however, it would be helpful if a short description of the various algorithms used in the paper and their differences were included in the material and methods

Response to Reviewer 5.

We thank the reviewer for their complimentary comments and review of our paper and are delighted to hear that they believe our manuscript to be “of high importance”.

Minor remark:

Overall the study is well read and the flow of the results presented is easy to understand however, it would be helpful if a short description of the various algorithms used in the paper and their differences were included in the material and methods

Response. We have now expanded our Methods section to include short description of the algorithms and their differences.